



# SCUBIDO: a Bayesian modelling approach to reconstruct palaeoclimate from multivariate lake sediment data

Laura Boyall[1], Andrew C. Parnell[2], Paul Lincoln[1], Antti Ojala[3,4], Armand Hernández[5], Celia Martin-Puertas[1].

[1.] Department of Geography, Royal Holloway University of London, Egham, TW20 0EX, UK.

[2.] School of Mathematics and Statistics, University College Dublin, Ireland.

[3.] Department of Geography and Geology, University of Turku, FI-20014, Finland

[4.] Geological Survey of Finland, Vuorimiehentie 5, FI-02151 Espoo, Finland

[5.] GRICA Group, Centro Interdisciplinar de Química e Bioloxía (CICA), Faculty of Sciences, Universidade de Coruña, Coruña, Spain.

*Correspondence to:* Laura Boyall (Laura.Boyall.2016@live.rhul.ac.uk)

## Abstract

Quantification of proxy records obtained from geological archives is key for extending the observational record to estimate the rate, strength, and impact of past climate changes, but also to validate climate model simulations, improving future climate predictions. SCUBIDO (Simulating Climate Using Bayesian Inference with proxy Data Observations), is a new statistical model for reconstructing palaeoclimate variability and its uncertainty using Bayesian inference on multivariate non-biological proxy data. We have developed the model for annually laminated (varved) lake sediments as they provide a high-temporal resolution to reconstructions with precise chronologies. This model uses non-destructive X-Ray Fluorescence core scanning (XRF-CS) data (chemical elemental composition of the sediments) because it can provide multivariate proxy information at a near continuous, sub-mm resolution, and when applied to annually laminated (varved) lake sediments or sediments with high accumulation rates, the reconstructions can be of an annual resolution.

SCUBIDO uses a calibration period of instrumental climate data and overlapping XRF-CS data to learn about the direct relationship between each geochemical element (reflecting different depositional processes) and climate, but also the covariant response between the elements and climate. The understanding of these relationships is then applied down core to transform the qualitative proxy data into a posterior distribution of palaeoclimate with



quantified uncertainties. In this paper, we describe the mathematical details of this Bayesian
approach and show detailed walk-through examples that reconstruct Holocene annual mean
temperature in central England and southern Finland. The mathematical details and code have
been synthesised into the R package SCUBIDO to encourage others to use this modelling
approach. Whilst the model has been designed and tested on varved sediments, XRF-CS data
from other types of sediment records which record a climate signal could also benefit from this
approach.

## 1.0 Introduction

Anthropogenic climate change over the most recent decades have enhanced the need to look
beyond the instrumental period to find common patterns to both today's climate and future
climate projections (IPCC, 2023; Kaufman and McKay, 2022). This calls for chronologically
constrained, climate-sensitive proxy records to extend the understanding of climate variability
beyond the instrumental period. These reconstructions can be used to contextualise present
changes observed in the climate system, identify recurrent trends which are unable to be
observed in the short instrumental record (e.g. decadal-centennial variability), and be used as
potential analogues for future climate scenarios (Bova et al., 2021; Liu et al., 2020; Snyder,
2010). In addition, quantitative reconstructions provide the opportunity to perform climate
sensitivity experiments between proxy reconstructions and climate model simulations,
strengthening climate projections for the future (Kageyama et al., 2018; Burls and Sagoo, 2022;
Zhu et al., 2022).

The Holocene Epoch (11,700 years to present) has been the focus of many proxy and

modelling investigations (e.g. Liu et al., 2014; Bader et al., 2020; Kaufman et al., 2020a; Bova
et al., 2021; Erb et al., 2022). This time period experienced temperatures which were similar
to today, and the availability of proxy records makes the Holocene a favourable interglacial to
investigate climate variability across multi-millennial timescales. Recently, there have been a
number of new reconstructions of global temperature which are based on large proxy dataset
compilations (Kaufman et al., 2020a; Kaufman et al., 2020b; Osman et al., 2021; Erb et al.,
2022). These synthesise different marine (Osman et al., 2021), or a combination of terrestrial
and marine (Kaufman et al., 2020b) proxy records and either use statistical approaches
(Kaufman et al., 2020a) or combine these with data assimilation (Osman et al., 2021; Erb et
al., 2022) to reconstruct climate both spatially and temporally. These have provided great
insight into climate variability across large spatial scales, of which are not possible when
looking at individual site records. However, they all have a common limitation which is the



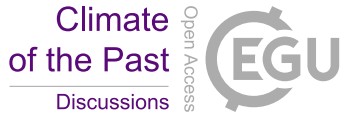

temporal resolution of their reconstructions. Due to the nature of the proxies included in the
large datasets (e.g. pollen, isotopes, foraminiera), the proxy signal is often non-continuous
creating a median reconstuction resolution of ca. 100-200 years (Kaufman et al., 2020b). Whilst
this temporal resolution is acceptable to look at spatially extensive and long-term climate
variability across centennial to millennial timescales (Cartapanis et al., 2022), higher frequency
variability such as the multi-decadal climate system is unable to be investigated, which is key
to improve climate predictions in this century (Cassou et al., 2018). Erb et al. (2023) used a
data assimilation approach which allowed them to upscale their temporal resolution to decadal.
However, this was only possible by including transient climate simulations, meaning that much
of the decadal climate variability observed in this reconstruction would be forced by the model,
rather than the proxy data itself given that only 11 out of the 1276 records have a decadal, or
higher temporal resolution.
Reconstructions of climate from a proxy record, whether this be a single-site, or a
compilation of multiple sites, require a transformation from the qualitative proxy value to a
quantified climate parameter with physical units of measurements (i.e. °C, mm of precipitation)
(Chevalier et al., 2020). A number of statistical or mechanistic methods can be used, each with
varying levels of complexity, uncertainties, and functionality (Tingley et al., 2012). Each
method requires a calibration stage or training set relying on modern observations of the
relationship between the proxy and climate which is then projected onto the proxy data (Juggins
and Birks, 2012). Quantitative approaches have matured from rather simplistic methods e.g.
linear regression (e.g. Imbrie and Kipp, 1971), to methods of increased complexity such as
weighted averaging regression (e.g. ter Braak and Juggins, 1993; Liu et al., 2020), composite
plus scaling (e.g. Jones et al., 2009; Kaufman et al., 2020a), modern analogue techniques (e.g.
Jiang et al., 2010), and artificial neural networks (e.g. Wegmann and Juame-Santero, 2023)
which are summarised well in Chevalier et al. (2020). Uncertain chronologies, assumptions in
proxy formation and preservation, and non-stationary relationships between the climate system
and proxy response through time are typical for many proxy records, which means that
interpreting the palaeo record has several complexities (Sweeney et al., 2018; Cahill et al.,
2023). Because of this, there has been a call for a greater reliance on hierarchical statistical
approaches, such as Bayesian statistics to reconstruct climate through time (Tingley et al.,
2012).

Bayesian statistics is an approach based on Bayes' Theorem and can be summarised as
applying prior knowledge to update the probability of a hypothesis when new data becomes





available (van de Schoot et al., 2021). It has been used to answer many statistical problems
which has included reconstructing palaeoclimate (e.g. Haslett et al., 2006; Parnell et al., 2015;
Tierney et al., 2019; Cahill et al., 2023). Many frequentist (non-Bayesian) approaches to
reconstructing climate mentioned previously often struggle to capture the complex
relationships inherent between climate and proxy data. This commonly occurs when the learnt
relationship in the calibration interval or training data is fixed, and then applied directly onto
the palaeo data which results in the assumption of a stationary relationship through time, and
fixed uncertainty estimates (Birks et al., 2012; Sweeney et al., 2018; Zander et al., 2024).
However, we argue that climate often exhibits non-stationary behaviour and this needs to be
captured in the chosen model. By contrast, a Bayesian approach allows a continued update
about the belief of the relationship between the proxy, the climate, and associated parameters
(Chu and Zhao, 2011). In addition, Bayesian analysis can holistically account for different
sources of uncertainty influencing a reconstruction (Birks et al., 2012; Sweeney et al., 2017).
Bayesian methods can consider the uncertainties at all stages of the modelling process and
model these as joint probability distributions producing properly quantified uncertainties with
credible intervals (Tingley and Huybers, 2010; Sweeney et al., 2018; Cahill et al., 2023).

A rising number of studies have used a Bayesian framework in their climate

reconstructions (e.g. Haslett et al., 2006; Holmström et al., 2015; Parnell et al., 2015; Tierney
et al., 2019; Hernández et al., 2020; Cahill et al., 2023). However, they provide low temporal
resolutions as they are based on non-continuously sampled proxies, resulting in reconstructions
of climate across multi-decadal to centennial timescales. This calls for a greater number of
quantified climate reconstructions using hierarchical modelling from records with refined
chronologies and proxies sampled at a high resolution.

Micro X-ray Fluorescence core scanning (XRF-CS hereafter) is a non-destructive

approach which provides qualitative multivariate information about the geochemical
composition of marine and lacustrine sediment cores (Davies et al., 2015). Sediment sequences
are continuously scanned enabling the proxy data to be produced at very high sampling
resolutions (0.2 mm). When this approach is applied on sediment sequences with either
sufficient sedimentation rates (>0.5 mm per year) or annual laminations (varves) (Zolitschka
et al., 2015), it can provide proxy information at a seasonal to decadal timescale. XRF-CS has
mostly been used to qualitatively reconstruct palaeoenvironments, as the relative changes in
geochemical composition of sediments are a direct response to the changing climatic and
environmental conditions in the lake-catchment system (Peti and Augustinus, 2022).



Our main goal here is to combine the advantages of using Bayesian inference in climate
reconstructions with the palaeoclimate value of varved records. In this methods-based paper
we aim to i) present a Bayesian approach to transform multivariate XRF-CS data into a
quantitative palaeoclimate dataset, ii) demonstrate the applicability of this approach on
different varved lake records from Europe, iii) compare the output of the Bayesian model to
previously published reconstructions to test the climatic reliability, and iv) promote its use
through the user-friendly R package, SCUBIDO.

**2.0 Methods**
**2.1 Proxy data**
The modelling approach has been built for the use of XRF-CS data as the chosen proxy. Raw
XRF-CS data originates in the form of element intensities which is often non-linear to the
concentration of elements in the sediment and can also be affected by the sediment physical
properties, measurement time and sample geometry, therefore we use centred-log ratios (clr
hereafter) to mitigate against these problems (Aitchison, 1986; Tjallingii et al., 2007; Weltje
and Tjallingii, 2008; Weltje et al., 2015; Dunlea et al., 2020). In this approach we do not assume
that any element has a stronger relationship with climate thus we include all clr-transformed
elements.

**2.3 Bayesian framework**
For our quantitative reconstruction of climate given the XRF-CS proxy data, we use Bayesian
inference and base our framework on the modelling approach described in Parnell et al. (2015)
and Hernández et al. (2020). Below we outline the notation used throughout:
▪ $C$ is used to represent the value of the climate variable at each time point.
▪ We use $XRF_{ij}$ to indicate the central logged transformed XRF-CS data at each depth of
the sediment core ($i$) where $i = 1, ..., n$ depths. As the XRF-CS data is multivariate, $j$
reflects the number of different central log ratio transformed elements ($j =$
$1, ..., n$ elements).
▪ $t_i$ denotes the calibrated age ($t$) of each depth ($i$) in cal years BP (before present where
present refers to 1950). It is important to note that age uncertainty is not considered in
this modelling approach.



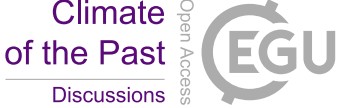

▪ $\theta$ is used to represent the parameters $(\mu, \beta_0, \beta_1, \beta_2)$ which govern the relationship
between each of the XRF-CS elements at each time point and the climate variable.
These are subscripted with $j$ to denote the element to which they refer.
▪ $\sigma_c$ is the standard deviation of climate per unit of time for our random walk model
detailed in this paper.
▪ A superscripted $m$ and $f$ are applied to each of the variables when referring to the
modern and fossil data sets respectively. For example, $C^m$ equates to the modern
climate, and $XRF^f$ refers to the fossil XRF-CS data.

The Bayesian posterior distribution we aim to calculate is outlined below:

(1)

$$p\big(C^f, \theta, \sigma_c \big| XRF^f, C^m, XRF^m\big) \propto p(XRF^m | C^m, \theta) \cdot p\big(XRF^f \big| C^f, \theta\big) \cdot p(C^f, C^m | \sigma_c)\, p(\sigma_c)\, p(\theta)$$

The posterior distribution on the left side of the equation $p\big(C^f, \theta, \sigma_c \big| XRF^f, C^m, XRF^m\big)$
represents the probability distribution of the fossil climate given fossil and modern XRF, and
modern climate. We use the likelihood expression $p(XRF^m | C^m, \theta)$ to represent the calibration
period where we learn about the relationship between the XRF-CS data and climate variable,
discussed in more detail in Sect. 2.3.2. $p\big(XRF^f \big| C^f, \theta\big)$ then represents the likelihood of the
fossil data given the climate, and finally $(C^f, C^m | \sigma_c)$ represents the prior distribution associated
with the fossil climate and its dynamics over time.

### 184 2.3.1 Model fitting

In order to fit the above model, we follow the computational shortcut of Parnell et al (2015)
which assumes that all the information about the calibration parameters (θ), comes from the
modern data. This means that the model is fit in two parts, with the first being the estimation
of θ within a calibration period, and then the second part which estimates the fossil climate
$(C^f)$ and $\sigma_c$. Thus, the resulting model becomes:

(2)

$$p\big(C^f, \theta, \sigma_c \big| XRF^f, C^m, XRF^m\big) \propto {\color{blue} p(\theta, \sigma_c | XRF^m, c^m)} \cdot {\color{blue} p\big(XRF^f \big| C^f, \theta, \sigma_c\big)} \cdot {\color{blue} p(C^f, C^m | \sigma_c)\, p(\sigma_c)}$$

Where the first term on the right-hand side (in blue) is estimated separately and
represents the posterior distribution of the modern calibration relationship parameters which is
then not further learnt from the fossil data in the second part of the model fit. Given the different



parts of the modelling approach, we split the following section into two, firstly fitting the
modern calibration period (Section 2.3.2), and then secondly using what is learnt from this
stage to reconstruct fossil climate (Section 2.3.3).

**2.3.2 Calibration model fitting**
Like all quantitative transformations of palaeoclimate, the first step is to understand the
relationship between the proxy and the climate variable. In our modelling approach this
relationship is learnt from the first term on the right-hand of equation 2 ($p(\theta, \sigma_c | XRF^m, c^m)$)
and includes not only the casual relationship between the individual XRF-CS elements and
climate, but also the covariance between the elements. The data used for this section of the
model is the most recent period and must be aligned with an overlapping period of instrumental
climate ($C^m$) and we call this our calibration dataset.
This step assumes that some of the variability observed in the proxy data is controlled
by the climate variable, this is sometimes referred to a 'forward' model. Here we want to
estimate the posterior distribution of the $\theta$ parameters ($\beta_0, \beta_1, \beta_2, \mu_0$) and the climate variability
parameter $\sigma_c$, from a joint probability distribution using the following:

(3)

$$p(\theta, \sigma_c | XRF^m, C^m) \propto p(XRF^m | C^m, \theta) \cdot p(C^m | \sigma_c) \cdot p(\theta)\, p(\sigma_c)$$

With $p(\theta)$ representing the prior distribution of the parameters $\beta_0, \beta_1, \beta_2, \mu_0$, with $\sigma_c$
and $p(C^m | \sigma_c)$ as the prior distribution on modern climate (we use a random walk with standard
deviation $\sigma_c$ at each time point). $p(XRF^m | C^m, \theta)$ is our likelihood distribution, and finally the
parameter's posterior distribution is represented by $p(\theta, \sigma_c | XRF^m, C^m)$.
To approximate the relationship between the clr-transformed XRF-CS data and the
climate, we use a multivariate normal polynomial regression model for each of the XRF
elements:

(4)

$$XRF_i^m \sim MVN(M_i, \Sigma)$$
$$M_i = [\mu_{i,1,\ldots,}\mu_{i,11}]$$
$$\mu_{ij} = \beta_{0j} + \beta_{1j} \cdot C(t_i) + \beta_{2j}\, C(t_i)^2$$





The mean term $\mu_{ij}$ captures the relationship between climate and assumes a quadratic
relationship with a single mode when $\beta_{2j} < 0$. We use $\Sigma$ to represent the covariance matrix of
the relationship between each of the different elements which are not explained by $\mu_{ij}$.
Vague normal distributions are used for the priors on $\beta_0$, $\beta_1$, and $\beta_2$, an inverse Wishart
prior on $\Sigma$, and finally a vague uniform prior distribution for $\sigma_c$:

(5)

$$B_{oj} \sim N(0,100), B_{1j} \sim N(0,100), B_{2j} \sim N(0,100)$$
$$\Sigma^{-1} \sim Wishart(R, k+1)$$

For the prior distribution on climate, we use a continuous time random walk:

(6)


$$P(C_i^m) \sim N(C_{i-1}^m, \omega_i)$$
$$\omega_i = (t_i^m - t_{i-1}^m) \cdot \sigma_c^2$$

Where $\sigma_c$ is also given a vague uniform distribution: $\sigma_c \sim U(0,100)$.

**2.3.3 Fossil model fitting**
Once the model has learnt about the relationship between the XRF-CS data and climate, the
second part of the computational shortcut can commence (Parnell et al., 2015). This first
involves using the learnt relationship to create marginal data posteriors (MDPs) which
represent all the information about fossil climate contained in one layer of XRF data. Thus, we
initially estimate the $C^f$ using only the information within a particular time slice ($XRF^f$). Using
only the information from one time slice at a time allows the model to marginalise over the
parameters ($\theta$) and reduce the dimensionality of the data. This step decreases the computational
burden of estimating both the climate - proxy relationship and the fossil climate values in the
same step. Information on the MDP fitting can be found in Supplementary Information 1 and
in more detail in Parnell et al. (2015; 2016).
To accurately capture the climate dynamics of the fossil period, we re-use the
continuous time random walk from the modern calibration module and combine each of the
individual MDP layers once they are corrected. This allows us to create a complete joint
posterior distribution of the combined $C^f$ and $C^m$ and fit the model detailed in equation 2. As
above, the varying time steps are captured via a dynamic precision term:





(7)


$$P(C_i^f) \sim N(C_{i-1}^f, \omega_i)$$

$$\omega_i = (t_i^f - t_{i-1}^f) \cdot \sigma_c^2$$


To fully learn the climate dynamics standard deviation parameter from both the fossil and the
modern data we set a log-normal prior distribution for $\sigma_c$:

(8)

$$\sigma_c \sim \text{LN}(a, b)$$


Where the values $a$ and $b$ are chosen to match the posterior distribution from the modern
calibration model fit.
The model produces an ensemble of posterior climate paths covering the fossil and
modern period. This takes into account the uncertainties in the XRF proxy climate relationship
with a mild smoothing constraint arising from the random walk prior. The ensemble can then
be summarised by taking the median value of the posterior distribution $C^f$ and calculating the
50% and 95% credible interval of the reconstruction using the 2.5%, 25%, 75%, and 97.5%
percentiles for plotting.

**Section 3.0 Walk through example**
This next section of the paper provides a walk-through example of each stage of the Bayesian
model fitting on real life XRF-CS data. In an attempt to make this modelling approach as user-
friendly as possible, we have produced the R package SCUBIDO (Simulating Climate Using
Bayesian Inference with proxy Data Observations) which synthesise the modelling process into
several distinct steps and can be downloaded from the GitHub repository:
https://github.com/LauraBoyall/SCUBIDO.
We demonstrate this example on the lake sediments of Diss Mere, a small lake in the
UK containing Holocene varved sediments. This site has been chosen due to the sediments
being annually laminated for much of the Holocene (from 2 to 10 thousand years before 1950
CE, cal. BP hereafter), and thus has a refined chronology based on annual layer counts with
age uncertainties of less than a few decades (Martin-Puertas et al., 2021). The averaged
sedimentation rate for the varved sequence is 0.4 mm/year with variability between 0.1 and 1.8
mm/year (Martin-Puertas et al., 2021). The most recent two millennia are recorded in the top





9 m of the sediment sequence, where the annual laminations are poorly preserved, and counting
was not possible. However, the chronology has been constrained through a series of radiometric
dating techniques ($^{14}$C, $^{137}$Cs) and tephrochronology, providing a high average sedimentation
rate of ca. 0.5 cm/year (Boyall et al., 2024). Both the modern sediment depositional processes,
and palaeo sediments have been studied in detail through modern lake monitoring, microfacies
analysis and analysis of the XRF-CS record, which all highlighted that the environmental
processes explaining the sediment deposition in the lake has not changed through time and
respond to climate variations on seasonal to multi-centennial timescales (Boyall et al., 2023;
Martin-Puertas et al., 2023; Boyall et al., 2024). Whilst human activity has had an impact on
the lake sedimentation in the last 2,000 years, i.e. increase the amount detrital input into the
lake (Boyall et al., 2024), the lake sedimentation and sediment composition keep responding
to the annual lake cycle (monomictic), which is driven by climate parameters such as
temperature and wind speed (Boyall et al., 2023). The sensitivity of these sediments to weather
and climate variability thus provides scope for testing this modelling approach.

The Diss Mere sediments were scanned using an ITRAX XRF-Core scanner (Cox

Analytical Systems) at the GFZ-Potsdam and geochemical elements include Si, S, K, Ca, Ti,
V, Mn, Fe, Rb, Sr and Zr at 200 μm resolution (Boyall et al., 2024). Boyall et al. (2024) found
a qualitative link between the XRF-CS data, specifically the element calcium (Ca) (linked to
temperature-induced authigenic calcite precipitation deposited during spring to early Autumn),
and annual mean temperature evolution through the Holocene (Davis et al., 2003; Kaufman et
al., 2020a; Rasmussen et al., 2007). Whilst this study found the strongest relationship to climate
with Ca, all the elements are used in this modelling approach given that SCUBIDO models the
covariance between the elements as well and learns from these relationships. For the first two
thousand years of the geochemical record between 10,300 cal a BP and 8,100 cal a BP, the
environmental interpretation of the element data reflected a non-climate, local signal associated
with the stabilisation of the lake depositional environment during the early Holocene (Boyall
et al., 2024). As a result of these findings, we attempt this modelling approach on only the
geochemical data from 8,100 cal a BP to present and emphasise to future users of SCUBIDO
that they must also conduct a qualitative analysis of the XRF-CS data and environmental
interpretation before using the model presented in this paper to investigate if their record is
climate sensitive.





## 3.1 Data set up

One of the most fundamental considerations for any type of palaeoclimate reconstruction is the choice of climate variable to reconstruct (e.g. annual mean temperatures, precipitation, growing season) given that different proxies are sensitive to a number of climate drivers (Sweeney et al., 2017). The SCUBIDO modelling approach can be easily adapted to reconstruct different climate parameters with overlapping instrumental data. However, it is important to note that not all lakes are responsive to every climate parameter of interest and thus the outputs may not be useful. For example, we attempted to run SCUBIDO on the Diss Mere XRF-CS data to reconstruct both temperature and precipitation, however the correlations between instrumental precipitation and individual elements were low and thus the model did not find a good enough relationship. Annual mean temperature on the other hand worked well, which support the temperature signal recorded in the qualitative XRF-CS data during the Holocene (Boyall et al., 2024). Another point to highlight at this stage is that we run the Bayesian model using a multivariate dataset made of the elements measured by the XRF scanner, which differentiate SCUBIDO from other recent reconstructions based on varved sediments (Zander et al., 2024). We do so to avoid any bias through time as the climate-proxy relationship might not be stable over time. SCUBIDO also includes the relationship between elements (covariance) to deal with this issue. As the top of the XRF-CS data (most recent period of sediment accumulation) begins at 1932 CE, a long-term instrumental temperature data set was required to get a sufficient length for the model to learn about the climate - proxy relationship. We therefore rely on the Hadley Central England Temperature (HadCET, Met Office) data which has been collecting temperature data since 1659 CE.

The first step was to divide the data into two: the modern calibration dataset (containing an age index ($t$), modern XRF-CS data ($XRF^m$) and the overlapping instrumental climate data ($C^m$)), and then the fossil data (containing the age ($t$) and XRF-CS data for the remaining data ($XRF^f$)). $XRF^m$ was resampled to annual means and was aligned with the corresponding year in the HadCET dataset. Given the start of the HadCET dataset beginning at 1659 CE and the top of the XRF-CS data finishing at 1932 CE, and a short gap where there was no XRF-CS data present, it meant that the calibration dataset was 290 years long. Temperatures were converted into anomalies from the mean of the calibration period as this not only removes the arbitrary mean of the temperature reconstruction making the data more comparable, but it can also better constrain the climate values in which the model picks from (see Supplementary Information 1). The fossil data was provided in its original temporal resolution ranging between





5 data points per year to >25 data points per year depending on the sediment accumulation rate.
This resulted in 56,069 time slices covering the period between 8,100 cal a BP and 1658 CE.

We check the model convergence using $\hat{R}$ values (Gelman and Rubin., 1992; Brooks

and Gelman., 1998) and evaluate the performance of the model using both in sample and out
of sample posterior predictive calibration checks (Gelman et al., 2008). We detail this analysis
in more detail below.

**3.2 Model fitting**
The full model was fitted using within the SCUBIDO R package. This package depends on
JAGS (Just Another Gibbs Sampler, Plummer, 2003) through the R package 'R2jags' (Su and
Yajima, 2021) to fit the modern calibration model and part of the fossil modelling stage. We
ran the calibration model for 100,000 iterations with a burn-in period of 40,000 and used a total
of 4 chains. The $\hat{R}$ values were consistently <1.05 indicating that the algorithm had successfully
converged during the Markov Chain Monte Carlo (MCMC) process (Vehtari et al., 2021; Su
and Yajima, 2021). Fig. 1 shows the quadratic relationships between the individual XRF-CS
elements and temperature in the calibration period.

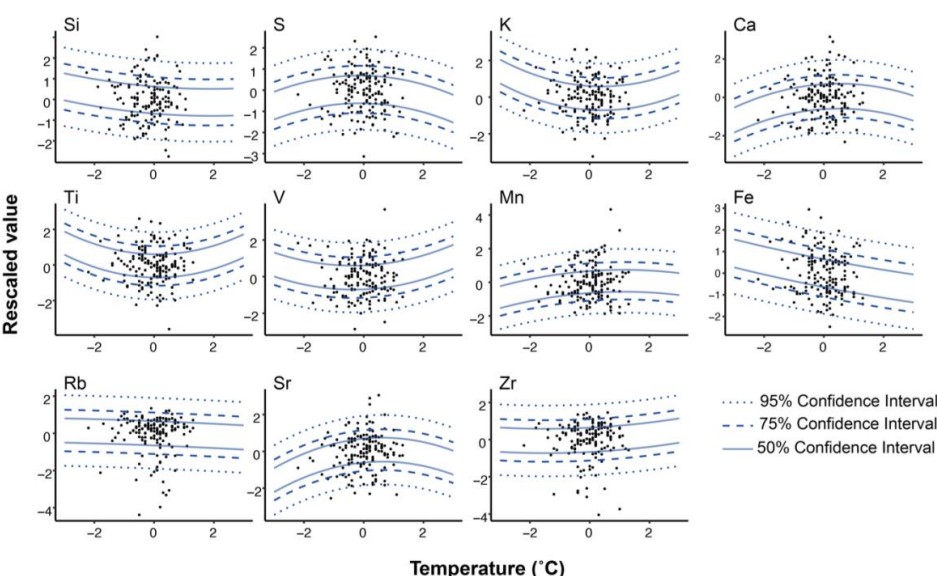

Figure 1: Relationship between the XRF-CS elements and instrumental annual mean temperature from the calibration period. Individual XRF-CS elements plotted against the instrumental climate anomaly data for each year. The quadratic relationships are represented by the lines with the solid lines representing the uncertainty ranges of 50%, 95% (dotted), 75% (dashed).




In more conventional approaches where XRF-CS data is used to qualitatively
reconstruct climate, only one element, or pair of elements (in the form of a ratio) is used at a
time to reconstruct climate (for example Zander et al., 2024). This would be equivalent to our
approach if had we used a diagonal structure for $\sum$ (equation 4). Such a diagonal structure treats
every element as independent and therefore may falsely reduce the uncertainty in the resulting
reconstructions. However, the novel contribution of our model is that it includes a multivariate
response regression approach that also models the covariances between the elements, and so
we argue produces more realistic, but also more uncertain reconstructions.
The fossil reconstruction stage for Diss Mere used 2,000 iterations with a burn-in period
of 200 with a total of 4 chains. Fewer iterations are required for this stage for convergence as
the model complexity is substantially reduced compared to the modern calibration stage as
MDPs are used. $\hat{R}$ values were <1.05 indicating satisfactory convergence of the algorithm. The
full reconstruction using all the SCUBIDO functions took approximately 16 hours on a
standard computer using a single core.

**3.3 Model validation**
As a more rigorous test of the model performance, we further test its uncertainty calibration
properties using an out of sample five-fold cross validation routine. Thus, we remove 20% of
the modern data and re-fit the full model to obtain posterior estimates of the climate variable
for years which the model has not seen during the training phase. We repeat this step five times
such that each observation year is removed once. We can compare these out of sample predicted
climate values with the true values in the modern data and see how often their uncertainty
ranges cross with the true values. For example, in an ideal model 95% of these values would
lie within the 95% interval and 50% in the 50% interval etc. Though in real-world data, the
estimated proportion inside the credible intervals may be slightly higher or lower. Out of
sample evaluation of climate reconstructions seems not to be a common feature in the literature
but we would strongly advocate this in the future.
The results of the five-fold cross validation showed that in 80% of the 199 calibration
temperatures, the reconstructions fell within the 95% credible interval (Fig. 2). The coverage
percentage for each individual fold ranged by 13%, from 75% to 88%. Given we are comparing
proxy data that are also affected by non-climate factors in the lake, the nature of the high
resolution (5-25 data points per year) XRF-CS data and the anomalous temperatures recorded
in the HadCET meteorological dataset, it is not surprising that the reconstruction does not



accurately reconstruct temperature within the 95% credible intervals, 95% of the time. In
addition, given that the calibration period occurs in the non-varved sediments where the
chronology has higher uncertainty (Boyall et al., 2024), it could mean that the XRF-CS data is
not perfectly aligned with the correct instrumental temperature thus lowering the validation
scores. On the other hand, the lower coverage percentage may also arise from the choice of
instrumental temperature data used in the calibration period as the temperatures are more
regional, whereas the μXRF-CS proxy data will be recording a local climate signal. In addition,
the earliest years of the HadCET dataset, the temperatures were based on non-instrumental
descriptions of weather and thus also subject to large uncertainties (Parker et al., 2010).
Nevertheless, gaining an 80% coverage percentage is acceptable for this modelling approach.

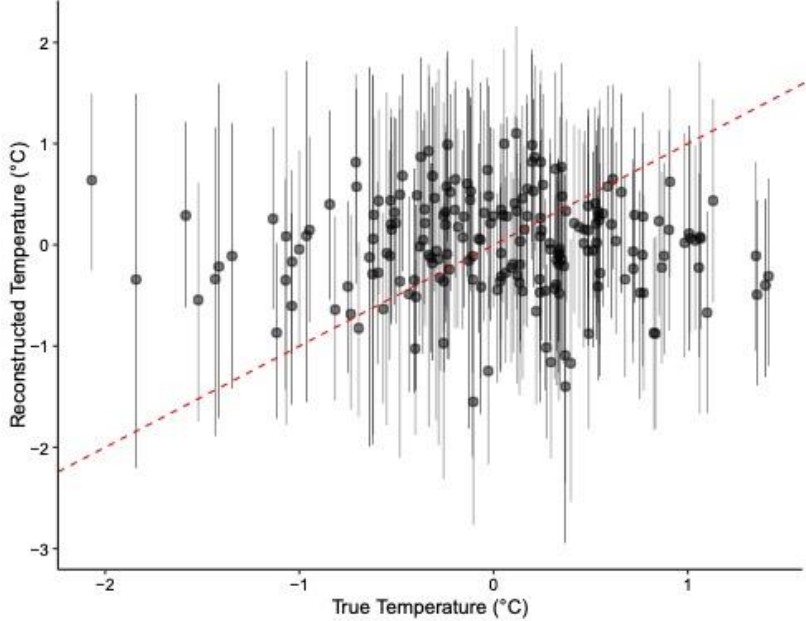

Figure 2: Results from the out of sample validation with true instrumental temperatures and reconstructed temperatures. Black dots represent the temperature values and error bars represent the predicted temperature's 95% uncertainty interval.






**Section 4.0 Annually resolved annual mean temperature reconstructions in Europe**

**4.1 Case site 1: Diss Mere, Central England**

The reconstruction of annually resolved temperatures for the past 8,100 cal a BP given the XRF-CS from Diss Mere using Bayesian inference is presented in Fig. 3. The median Holocene temperature reconstructed from Diss Mere is 9.65 °C and has a maximum range of 1.97 °C with temperature anomalies between -1.50 °C and 0.49 °C (7.66°C and 9.65 °C absolute temperatures). Most of the temperatures before ca. 2,000 cal a BP are cooler than present (9.16 °C) with only isolated centennial-scale periods where temperatures are warmer (Fig. 3). Inclusive of the credible intervals, the reconstructed Holocene variance is slightly greater than the instrumental period with a standard deviation of 0.63 °C for the reconstruction and 0.61 °C for the HadCET instrumental temperature. The centennial to interannual variability is, however, reduced in the last two millennia, similar to present time variability. The first millennium of the common era is slightly warmer than today remaining similar to present (Fig. 3).

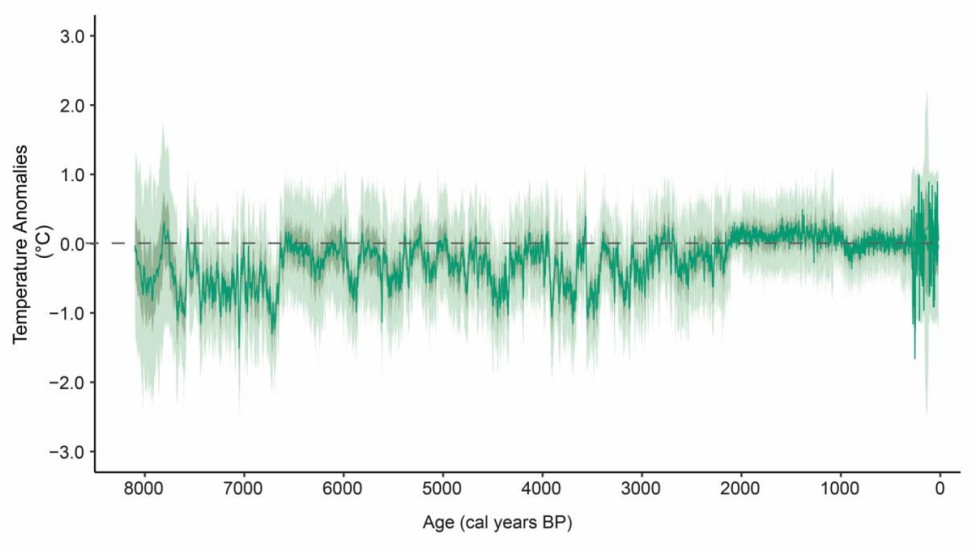

Figure 3: Annually resolved temperature reconstruction from Diss Mere. Dark green line represents the median reconstruction with 50[th] percentile and 95[th] percentile in darker green and light green, respectively. The data is presented in anomalies for the UK long-term average 1991-2020 and the dashed grey line marks the centred mean of 0 °C using this period.



## 4.2 Case site 2: Lake Nautajärvi, Southern Finland

We have applied the SCUBIDO approach to reconstruct Holocene annual mean temperature from Nautajärvi, a lake in southern Finland with a different stratigraphy to Diss Mere. Lake Nautajärvi is also a varved lake but shows an uninterrupted laminated sediment from the early Holocene to present (Ojala and Alenius, 2005). Except for the first 200 years of the record (9,852 – 9,625 cal a BP) when varves are thick (ca. 5 mm) due to a high detrital input during the formation of the lake (Ojala and Alenius, 2005; Ojala et al., 2008b), the sedimentation rate (0.2 – 1.6 mm/year) is similar to the varve thickness of Diss Mere (0.1 – 1.4 mm/year). Analysis of both the sediments and the XRF-CS data from Nautajärvi revealed that the lake, and subsequent sediment record is responsive to climate variability (Ojala et al., 2008a; Lincoln et al. in review) thus is a good record to also apply this Bayesian methodology on. Table 1 summarises the characteristics of the modelling approach applied on lake Nautajärvi varved sediment sequence.

Table 1. Summary table of the Lake Nautajärvi data used for the Bayesian reconstruction.

| | | |
|---|---|---|
| **XRF-CS details** | XRF-CS set up | |
| | XRF-CS elements used | Al, Si, S, K, Ca, Ti, V, Cr, Mn, Fe, Cu, Rb, Sr, and Zr |
| **Calibration data** | Meteorological data | Temperature data for Nautajärvi was from 16 weather stations within a 200 km radius from the lake obtained gathered using the 'rnoaa' package (Chamberlain et al., 2024). Annual mean temperature is used. Data preservation from the interwar years (1918-1945) is limited and/or missing thus these have been excluded from the calibration dataset (Supplementary Figure 1) |
| | Age range | -70 to 68 cal a BP |
| | Number of time slices | 102 |
| **Reconstruction data** | Age range | 69 to 9829 cal a BP |
| | Number of time slices | 16418 |

Figure 4 shows the annual temperature reconstruction from Nautajärvi for the past ca. 9,800 years overlaid on top of the Diss Mere reconstruction. The median Holocene temperature reconstructed from Nautajärvi is 5.1 °C (Supplementary Figure 3) and had a range of 1.60 °C




between 4.22 °C and 6.03 °C (-0.39 °C and 1.22 °C, anomalies) which is within the range of
variability observed during the instrumental period. Overall, the reconstructed Holocene
temperatures at Nautajärvi is cooler than except for the period between ca. 7,000 and 4,000 cal
a BP where temperatures are warmer and have the highest Holocene variance.

The comparison of Nautajärvi and Diss Mere through the Holocene shows slightly

different multi-millennial temperature evolutions where temperatures in England steadily
increase whereas Finland reaches maximum temperatures in the mid-Holocene and then
decreases thereafter (Fig. 4). We discuss millennial-scale trends in the next section when we
compare our reconstructions with published low-resolution Holocene temperature
reconstructions. On multi-decadal to centennial timescales, there is a good agreement between
the anomaly value reconstructions at both sites showing similar trends and amplitude of
change, especially on variability during the mid-Holocene from ca. 4,000 to 6,500 cal, yr BP
(Supplementary Figure 4). Larger variability in Diss Mere (England) prior to 6,500 cal yr BP
compared to Nautajärvi (Finland) might be reflecting different regional climate sensitivity
during a period when the instability of the Laurentide ice sheet and hydrological changes in the
Baltic Sea region was still having an important role on the reconfiguration of the climate system
and spatial distribution of climate patterns in the Northern Hemisphere (Yu and Harrison, 1995;
Wastegård, 2022).

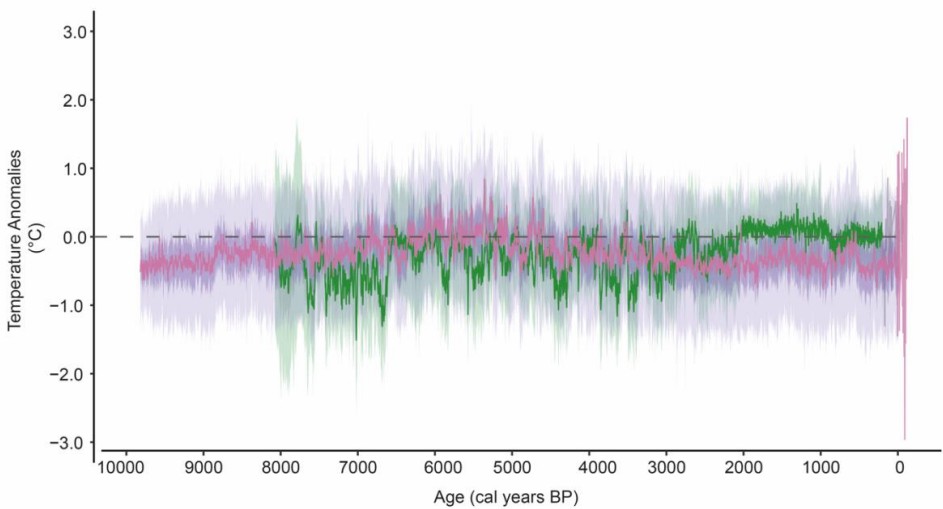

Figure 4. Annually resolved temperature reconstruction from Nautajärvi for the past ca 9,800 years overlaid on Diss Mere's reconstruction. Dark pink line represents the median reconstruction with 50th percentile and 95th percentile in darker purple and light purple, respectively. The anomalies are calculated with reference to the 1991-2020 mean from the instrumental data. The grey dashed line marks the 0 °C mean.




### 4.3 Palaeoclimate comparisons


To test whether the temperatures produced from the SCUBIDO modelling approach are
sensible on longer timescales, we compare our results from Diss Mere and Nautajärvi with
previously published proxy reconstructions (Temp12k, Kaufman et al., 2020a) and data
assimilation results (LGMR, Osman et al., 2021; Holocene-DA, Erb et al., 2022) for the same
period (Fig. 5). We choose these reconstructions to compare with because they are all based on
large-scale data compilations utilising a range of models and proxy types. The Temp12k and
Holocene-DA reconstructions both use the Temperature 12k proxy database (Kaufman et al.,
2020b) with the Temp-12k reconstruction using a multi-method ensemble to reconstruct
temperatures at a centennial resolution (Kaufman et al., 2020a) and the Holocene-DA using an
updated version of this dataset in a data assimilation framework to combine with transient
climate simulations in order to get a reconstruction of temperature at a decadal resolution (Erb
et al., 2022). On the other hand, the LGMR reconstruction uses only marine proxy records in a
data assimilation approach to produce a reconstruction of temperature at a multi-centennial
resolution.
The multi-millennial trends in the reconstructions are best demonstrated with both Fig.
5a and b showing the clear evolution of temperatures through the Holocene. Fig. 5a shows the
slope from linear models conducted on the different reconstructions to explore the evolution of
temperature through time. The Diss Mere, Holocene-DA (Erb et al., 2022), and LGMR (Last
Glacial Maximum Reanalysis, Osman et al., 2021) linear models all demonstrate an
amelioration of temperature through the Holocene with similar rates of warming, especially
during the mid-Holocene where there are almost no differences between the records (Fig. 5a).
The Temp-12k reconstruction from Kaufman et al. (2020a) and the Nautajärvi reconstruction
from this study deviate from the general increasing trend observed in the other reconstructions
and instead show an overall decrease in temperature from the early to late Holocene (Fig. 5a).
These records have a more definitive early Holocene Thermal Maximum (HTM) with cooling
thereafter in comparison with the other reconstructions, hence the linear model describing a
general decrease in temperature through time. As part of the current discussion on the Holocene
temperature conundrum (Liu et al., 2020), the differences in temperature evolution between the
reconstructions may be a factor of a seasonal bias, which has been already noted for the Temp-
12k reconstruction reflecting mostly summer conditions and/or spatial imbalances in proxy
distributions (Bova et al., 2021; Erb et al., 2022).





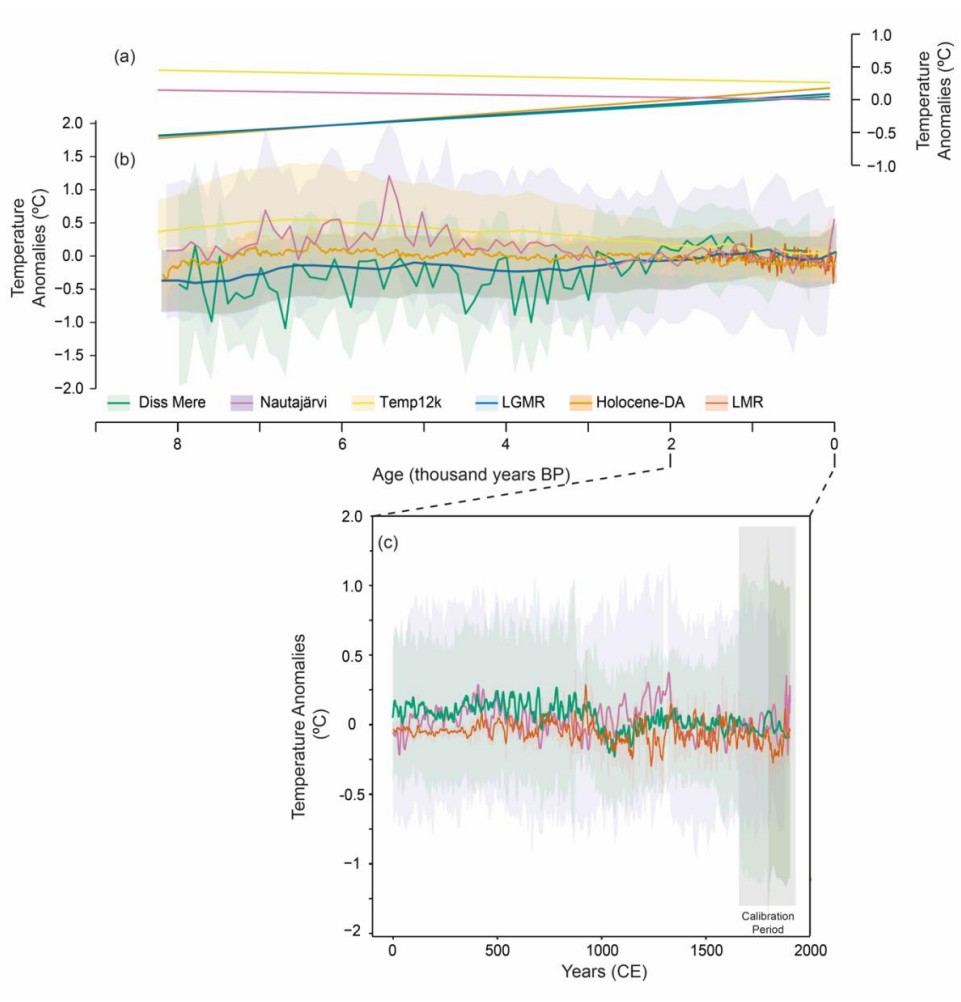

Figure 5: Comparison between different Holocene temperature reconstructions in anomalies. Note that the reference period for all these reconstructions is the mean between 2000 to 0 cal a BP. a) linear relationships between the reconstructed temperature and time for Diss Mere (green) Nautajärvi (purple), LGMR (Osman et al., 2021) (blue), Temp12k (Kaufman et al., 2020) (Yellow) and the Holocene-DA (Erb et al., 2023) (orange). b) The reconstructions from the above studies with Diss Mere and Nautajärvi resampled to 100 years to explore the centennial scale variability and match the resolution of the other reconstructions. The LGMR and Temp12k presented at a 200-year. The envelopes for each line in the respective colours represent the uncertainty for each reconstruction. c) a focus window on the common era with the Diss Mere temperature reconstruction with the LMR (Tardif et al., 2019) (orange) for a grid 5°W:15°E, 50:60°N. The solid bold lines are at 10-year decadal moving average whereas the transparent envelopes are the original annual resolution.


The amplitude of variability from the SCUBIDO-produced reconstructions from this

study is much larger than the global reconstructions. Ultimately this is because the LGMR and
Temp12k have low temporal resolutions causing the reconstruction to be smoothed, and also
contains a range of proxy types. Whilst the Holocene-DA reconstruction technically has a data



every 10 years, as mentioned in their study, the reconstruction does not contain robust decadal
information from the proxy records and is achieved instead by utilising both proxy and transient
models together and thus the low amplitude is still inherent from the low-resolution proxy data
used.

**4.3.1 The last two millennia**
Reconstructing palaeoclimate for the common era (past 2,000 years) has been the focus of
many climate studies (e.g. Smerdon and Pollack, 2016; PAGES2k Consortium, 2017a; Tardif
et al., 2019; Anchukaitis and Smerdon, 2022). To test the Bayesian reconstructions from this
study through a period of increased anthropogenic disturbance, we compare the reconstructions
to the Last Millennium Reanalysis (LMR, Tardif et al., 2019) (Fig. 5c). Whilst the LMR and
the Bayesian reconstructions are annual, we decide to compare at a 10-year resolution to reduce
noise and explore the main decadal-scale trends between each record.  Despite increased
anthropogenic disturbance to the lake system over the past 2,000 years at Diss Mere (Boyall et
al., 2024), and a disruption to the proxy signal and lake functioning, the comparison between
the overall trend of the LMR and Bayesian temperature reconstructions are good, especially at
Diss Mere (Fig. 5c). Correlation coefficients between the LMR and Diss Mere is $r = 0.58$, P =
<0.0001, however no statistically significant correlations could be made between Nautajärvi
and the LMR despite the general similar evolution trend in Fig. 5c.
In the first millennia (0-1000 CE), the LMR is much less variable than the Bayesian
reconstructions, with slightly cooler temperatures and negative anomalies (Fig. 5c). The lower
variability in the LMR is probably attributed to the very low number of proxy records used for
the first few hundred years of the reconstruction (Tardif et al., 2019). Despite the minor
differences in the amplitude of variability, each record shows a warmer first millennium
compared to the second, which has been discussed in previous reconstructions (PAGES
Consortium, 2017b; Esper et al., 2024). Once the decrease in temperature occurs at ca. 850 CE
at Diss Mere and LMR and 1200 CE at Nautajärvi, there is a better agreement in both the
temperatures and amplitude of variability until present (Fig. 5c) resulting in a better agreement
between these records than the previous millennium. The consistency between the records
highlights that despite the different sediment varve characteristics, varve formation processes,
and interactions between sedimentation and human activity, the Bayesian approach is able to
reconstruct a quantified, local to regional climate record from the XRF-CS.





**5.0 Conclusions and recommendations for future use of SCUBIDO**

This study presents the first attempt at reconstructing quantitative annual mean temperatures from multivariate XRF-data from sediment records using Bayesian inference. Several methodological decisions were made when building SCUBIDO which we believe can help contribute to the advancement of climate reconstructions. The most important choice was to use of Bayesian inference to not only get a single temperature estimate at each time point, but to also get a full posterior distribution to properly quantify uncertainties. In addition, we designed the model to include all geochemical elements and have SCUBIDO model their covariances instead of relying on prior assumptions about relationships, and the final choice was to synthesise SCUBDIO into an R package for the community. We believe that this was the best way to be as user friendly as possible as we think others could find this approach interesting and help make new annually resolved palaeoclimate reconstructions.

The ability of Bayesian in handing various types of data, changing timesteps/resolutions, and gaps within datasets has been utilised in this study, for example, there are periods within both the XRF-CS records from Diss Mere and Nautajärvi which have short gaps and periods where the sedimentation rates are variable resulting in changing time steps. However, this was easily mitigated against by using a Bayesian framework.

In this paper we apply SCUBIDO to two proxy records to reconstruct Holocene annual mean temperature in Europe and the results showed consistency with previously published paleoclimate reconstructions on a multi-millennial timescale. However, given the model and the high-resolution proxy data from this study it provides a much more detailed overview of temperature evolution through the Holocene by increasing the resolution to annual at a single site. Of course, the records we compared to (Holocen-DA, Temp12k, and LGMR) have the advantage of also being spatial reconstructions and not just temporal like in our study. The goal would be for more people in the palaeoclimate community to use SCUBIDO and thus produce more reconstructions of an annual resolution to then be incorporated into these large data compilations.

Whilst we encourage other groups to use this approach on their XRF-CS records, there are some precautions which should be taken since SCUBIDO does not provide a physical model between the climate and geochemical sediment composition. Like all palaeoclimate reconstructions using different statistical techniques, there is still some assumption that the proxy-climate relationship does not deviate too much through time to what is observed in the calibration period. This is important to consider when sites have experienced substantial alterations in human activity or other depositional changes, and we recommend to carefully

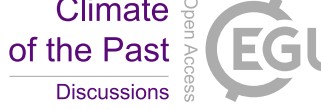

check that the major shifts in the climate reconstruction are explained from climate or rather
be explained by changes in the sedimentology (e.g. transitions from varved to non-varved
deposits and changes in the varve microfacies). Because of this, we encourage users to
qualitatively interpret the XRF-CS record to see whether the lake remains sensitive to climate
through time, as well as finding the climate parameter to which the lake is sensitive to. And
finally, because XRF-CS data is highly site-specific and sensitive to local systems, it is not
possible to calibrate one site and apply that calibration period on another XRF-CS lake record
which may be common in other proxies e.g. pollen (Parnell et al., 2016).
Future developments of the SUBIDO approach may include integrating age uncertainty
into the model as currently age ensembles are not used. This means that at present lake data
with stronger chronological age models would likely produce better reconstructions, as
aligning the calibration instrumental climate data with the correct layers of XRF-CS data is
important.

**Author contribution**

LB, AP, and AH, and CMP conceptualised the study. LB, AH, and AP created the methodology
and software, LB made the R package. LB, AP, PL, AH, and CMP were involved in the
discussion and formal analysis. CMP, PL, and AO were involved in data curation. LB wrote
the original manuscript with supervision from AP and CMP and all authors were involved in
the review and editing process.

**Competing interests**

The authors declare that they have no conflict of interest.

**Acknowledgements**

This study was funded by a UKRI Future Leaders Fellowship held by Celia Martin-Puertas and
contributes to the DECADAL project 'Rethinking Palaeoclimatology for Society' (ref:
MR/W009641/1) of which Paul Lincoln if funded by. Laura Boyall is funded by Royal
Holloway University of London through a PhD studentship. Andrew Parnell's work was
supported by Research Ireland Research Centre awards Climate+ 22/CC/11103 and Insight
12/RC/2289_P2. Armand Hernández is supported by the Spanish Ministry of Science and
Innovation through the Ramón y Cajal Scheme (RYC2020-029253-I). The authors thank Rik
Tjallingii for the provision of the XRF data and for comments on the manuscript.





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
