# Peer review of "SCUBIDO: a Bayesian modelling approach to reconstruct"

_Climate of the Past, 2024_

## Author Comment (AC1)

We would like to thank Pierre Francus (Reviewer 1) for his constructive suggestions and comments, which have helped to improve our manuscript entitled "SCUBIDO: a Bayesian modelling approach to reconstruct palaeoclimate from multivariate lake sediment data." Below, the reviewer's comments are shown in red, and our responses in black.

I read this article with great interest. It's very easy to read, although the part describing the Bayesian model is more difficult, probably because my understanding of these statistics is very limited. I'll leave it to others to comment on this mathematical part. The proposed approach should be of interest to several researchers active in the study of varves, and I'm looking forward to trying it out myself. Applying this approach to all varved sites should, as the authors write in their conclusion, « (…) produce more reconstructions of an annual resolution to then be incorporated into large data compilations ». It is therefore a significant contribution to paleoclimatology.

We thank you for your interest in our work. Like you, we hope that members of the varve and palaeolimnology community will find this research both interesting and useful.

Although the article is very interesting, I am very surprised that the authors do not show any comparison of their new Bayesian reconstructions with the existing reconstructions for Nautajarvii and Diss Mere. It seems to me that such comparisons would have been far more relevant than comparisons of the Bayesian reconstructions of these two sites with each other, or with reconstructions based on large-scale databases (Temp 12k, LGMR, Holocene-DA, and LMR). If the authors have good reasons for not making this comparison, it's essential to say why.

We thank the reviewer for this comment. Whilst there have been previous palaeoclimate reconstructions published from these lakes, neither lake has produced an annual mean temperature reconstruction per se.

**Diss Mere:** Martin-Puertas et al. (2023) presented varve thickness data as a proxy of climate variability and compared to a AMOC simulation to explore decadal oscillatory variability in the North Atlantic realm. The thickness of the summer laminae (authigenic calcite precipitation) was also compared to a summer temperature simulation for this region to support the response of the lake to a specific climate parameter. Based on these findings, Boyall et al. (2024) suggested that the Ca-clr record as a proxy for authigenic calcite precipitation, might be also responding to summer temperature as Ca is mostly deposited in the summer months, supported by the findings from a modern lake monitoring investigation at Diss Mere published in Boyall et al (2023). We decided not to include the comparison with the varve data or the Ca-clr profile in the main manuscript as we thought that it may lead to misunderstanding as these proxy records show a clear seasonal bias toward the summer and show a different temperature evolution (i.e. proxies sensitive to summer temperature show a Holocene Thermal Maximum while the SCUBIDO annual mean temperature shows gradual warming thorough the Holocene, see response to Reviewer 2 about the HTM). However, we attach here a comparison between the Diss Mere temperature reconstruction from this study and the Ca-clr record from Boyall et al. (2024) in case you are interested to see the differences between the two records.

[Figure]

Figure A1-1 Comparison between Diss Mere annual mean temperature reconstruction in green for this manuscript and the Ca-clr record from Diss Mere published in Boyall et al. (2024) in red. Both records are presented at a 10-year moving average.

**Nautajärvi**: similar to Diss Mere, varves are responding to a combination of different climatic parameters and disentangling them from the varved record is challenging (Ojala et al., 2005). There is a pollen-inferred growing degree days (GDD) reconstruction (Ojala et al., 2008), and whilst this tends to have a good relationship with annual mean temperature, GDD reflects variability in the growing season (summer) and can indicate both long, mild summers, or short and hot summers. The figure below is the Nautajärvi reconstruction from SCUBIDO overlaid with the GDD and whilst the long-term trend is similar, we prefer not to show this comparison within the manuscript given that they are reconstructions of two different climate parameters and thus may not add much to the discission in this current manuscript.

In the revised manuscript we will explain to interested readers why we have compared with the large dataset reconstructions rather than site-specific reconstructions.

[Figure]

Figure A1-2: Nautajärvi annual mean temperature reconstruction from this study (purple)with growing degrees reconstruction from Ojala et al. (2008) (red line)

Figures 1 and 2 show no relationship between XRF-CS or reconstructed with actual temperature, at least when considering a classical statistical approach. It's something of a surprise to see that a Bayesian approach manages to derive information from these relationships that are not visible. Could you address this in your discussion? It might help convince readers of the validity of your approach.

We agree with all three reviewers that Figure 1 does not demonstrate clear relationships between temperature and the individual XRF elements. However, we would like to emphasise that within this modelling approach we are not using the relatively weak relationship between each element and climate individually and instead are harvesting from the joint response of each of the elements together in a multivariate response regression approach (sometimes known as seemingly unrelated regression; SUR, for example Mbah et al. 2018) which provides us with a more precise posterior estimate of climate. This means that the model is learning from both the direct quadratic relationship between element and climate, and also the group response to climate represented as correlations between the elements. A similar example of this is when pollen is used to reconstruct climate. A single species may have a weak relationship with the climate target variable, but when used in combination with species assemblage it is this joint relationship that can find a good match with a climate variable. This is a very common approach in Bayesian reconstructions of palaeoclimate (e.g. Haslett et al 2006, Parnell et al 2015), though individual relationships are often not shown in papers. We have decided to keep Figure 1 as we believe some readers will be interested in observing the relationships between the individual elements and temperature. However, to ensure that this figure is not misinterpreted, we will include additional sentences to the main text and the figure captions explaining the reasons behind the weak looking relationships.

In response to the concerns about Figure 2, we agree that in the previous version of the manuscript the relationship between true and reconstructed climate was not illustrated well. Since this initial submission we have re-assessed the calibration period and have now started the calibration period at 1700 CE rather than 1669 CE as there were several gaps in the Diss Mere XRF data between 1659 CE and also documented uncertainties in the HadCET dataset. We have also identified in our script used for validation that there was a small coding bug which meant that we were misaligning our true instrumental temperature and reconstructed temperature by one year (this was due to us rounding up the decimal places in the ages). This has since been adjusted and a new Figure 2 will be presented in the manuscript which displays a good relationship between true and reconstructed temperature. But we also attach it here for you to view now (Figure A1-3), as well as the Nautajärvi plot (Figure A1-4) which will be going into the Supplementary Information.

[Figure]

Figure A2-3: The results from the out-of-sample cross validation results for Diss Mere with true temperature used in the calibration period against reconstructed temperature from the SCUBIDO model. Colours represent the different folds and lines represent the 95% confidence interval. This will be replaced as Figure 2 in the updated version of the manuscript.

[Figure]

Figure A1-4: The results from the out-of-sample cross validation results for Diss Mere with true temperature used in the calibration period against reconstructed temperature from the SCUBIDO model. Colours represent the different folds and lines represent the 95% confidence interval. This will be replaced as Figure 2 in the updated version of the manuscript.

If the upper part of Diss Mere contains no varved record, it would be desirable to better describe the chronology of the part that served as Modern calibration dataset between 1659 CE and 1932 CE. I suggest adding a supplement, but it seems important to me to have a figure so that the reader can appreciate the quality of the age model in this time interval.

The full chronology and age model for the non-varved sediments is described and published in Boyall et al. (2024). This is based on a combination of tephra layers that link the non-varved and the varve chronology, radiocarbon dates and the 1963 CE 137Cs peak. The average age uncertainty for the non-varved section of Diss Mere is ± 65 years and thus is higher than the varve chronology. During the calibration period (1700 – 1932 CE), the age uncertainty is smaller at the top of the sequence with a maximum uncertainty of ± 22 years between 1932 CE and 1800 CE, however this increases gradually to ± 110 years for the following century (Supplementary Figure 1). The sedimentation rate within this period is very high (0.15 cm/year) and includes up to 20 data points per year.

We agree with the reviewer that the information about the calibration period should be included in the supplementary information and thus we will add this, and a age-model for this period. This will go as Supplementary Figure 2 but is Figure A1-5 in this response.

[Figure]

Figure A1-5: Age mode for the calibration period (1700-1932 CE) for Diss Mere. This age model is part of the published age-model from Boyall et al. (2024).

Finally, it would be good to define all the variables in the equations presented. Some are indeed not defined, such as MVN, Mi, ωi,... Perhaps in a list of abbreviations and variables in an appendix?

This was a great idea, and we thank the reviewer for this. We have created this table and will add this as Supplementary Table 1.

I have further minor comments which are listed below.
L23: add « Micro » before X-Ray.

We will correct this in the revised manuscript.

L73: what is the starting temporal resolution Erb? Please specify in the text

We will correct this in the revised manuscript.

L136: replace "different" by "two"

We will correct this in the revised manuscript.

L148-149: CLR-transformation requires that elements having too many "0" values are dismissed. Have you done that?

Yes, only the elements which did not contain a large number of null values were included. We will clarify in the text when we introduce the Diss Mere data.

L307-309: along the same lines, how these elements have been selected? Maybe should you also specify the dwell time for each acquisition step.

We will clarify this in the text that they were chosen based on having a <15% standard error. We will also add in the dwell time.

L319: I suggest replacing "As a result of these findings" by "Therefore"

We will correct this in the revised manuscript.

L333: what temperature are you referring to here, because two lines further on, you say you're using the average annual temperature.

We will clarify that we mean annual mean temperature.

L338-339 : « which differentiate SCUBIDO from other recent reconstructions based on varved sediments (Zander et al., 2024)". This information is available later (L375-377) and more detailed there. I would delete this part of the sentence.

We will correct this in the revised manuscript.

L350: How have you resampled to annual means? Using the varve boundaries? Or using the equation of the age model?

We will clarify in the text that we have just used linear interpolation. We could not use the varve boundaries as this was in the uppermost non-varved section.

L367-370: Please try to use less jargon, or explain what a "Burn-in period", and the "chains" are.

We will correct this in the revised manuscript.

L392: a reference to the literature would be useful here.
We will correct this in the revised manuscript.

L414: please use "µXRF-CS" everywhere or nowhere.
We will correct this in the revised manuscript.

L431: it seems that the last 2000 years are less variable in Diss Mere because this corresponds to the non-varved section.
We also believe that this could be the case, though this happens slightly before the end of the varves and thus did not include this interpretation in the previous version. However, we will add this into the sentence as a potential cause for the loss of variability.

L437: Do you want to say "lithology" instead of "stratigraphy".
We will correct this in the revised manuscript.

Table 1 is missing some information, such as the how the elements have been selected, the dwell time, the resolution, the anode composition of the X-ray tube.
We had initially not included this as the information is present in Lincoln et al. (2025). But we will now add some of this information into Table 1 and will refer readers to Lincoln et al. (2025) for more detail.

Figure 4 "please specify in the caption that Nautajarvi is the pink curve and Diss Mere the green one.
We will correct this in the revised manuscript.

L554: "handling" instead of "handing"?
We will correct this in the revised manuscript.

Please add a section "Data availability" as required by the editorial policy. Reference to the codes (Gitub) and, if you can, the XRF-CS data. Maybe also indicate where the Temp 12k, LGMR, Holocene-DA, and LMR datasets have been downloaded from.
We will correct this in the revised manuscript.

Additional references not included in the manuscript bibliography but mentioned in the response:

Mbah, C., Peremans, K., Van Aelst, S., & Benoit, D. F. (2018). Robust Bayesian seemingly unrelated regression model. Computational Statistics, 34(3), 1135–1157. https://doi.org/10.1007/s00180-018-0823-x

---

## Author Comment (AC2)

We would like to thank Maarten Blaauw (Reviewer 2) for his constructive suggestions and comments, which have helped to improve our manuscript entitled "SCUBIDO: a Bayesian modelling approach to reconstruct palaeoclimate from multivariate lake sediment data." Below, the reviewer's comments are shown in red, and our responses in black.

Inferring fossil climate from calibrating XRF to recent temperature by necessity assumes that the relationship has been stationary over time, and that any other factors which didn't occur as much further back in time, e.g. soil erosion or nutrient pollution, do not affect the temperature reconstructions (or even more importantly, the calibration period). However, humans have messed up many records over the past centuries/millennium, possibly affecting the varved record as well (note that especially Diss Mere is located in a severely human-influenced region). How can these factors be disentangled? This is mentioned in the discussion, but it would be useful to expand more on this problem in the Introduction.

We completely agree with this reviewer that humans have changed the landscape over the last several thousand years. We believe that this is a problem faced by all climate reconstructions whether located in an urbanised area, such as Diss Mere, or more remote, such as Nautajärvi as the calibration period lies in a period where humans have already altered climate behaviour. Unfortunately, our model is not a physical model which can specifically disentangle the role of human activity, though potentially at the price of uncertainty quantification. This is why we emphasise that the user should carry out a preliminary investigation to explore the role of human activity to ensure that there are not significant changes to the geochemical signal. Whilst this information cannot be explicitly fed into the model, it can help with understanding whether the data should or should not be used in this approach. For the example of Diss Mere, Boyall et al (2024) found no significant changes in terms of what the different elements were responding to between 8,000 to present, and rather human activity just influenced the variability. So, whilst human activity likely had a role in the variability shift at ca. 2,000 we believe that the climate signal remains. We think this is a good idea to mention though as the reviewer suggested and thus we will add additional emphasis to the section in the conclusion to make users cautious of the results if there are significant changes to their µXRF-CS records, and will add more information about how human activity can lead to challenges in palaeoclimate reconstructions.

Why did you decide to only use non-biological proxy data? Perhaps this has to do with additional complexities of interacting/competing ecosystem components, but it would be good to spell this out (e.g., in section 2.1).

There are two main reasons: 1) the temporal resolution of the reconstruction and 2) potential seasonal and ecological biases.

The purpose of this approach is to reconstruct climate at an annual to decadal resolution. Because  $\mu$ XRF-CS data has a sample resolution <1 mm, and often as low as 0.2 mm, it means that, for many sediment records, we are able to get several data points each year. For example, in Diss Mere there are about 25 data points on average during the non-varved sections (sedimentation rate = 0.5 cm/yr) and 5 data points within the varved sections (0.4 mm/yr). Such a high sampling resolution for proxies

that require to take a sample of sediments (e.g. pollen, diatoms, chironomids) is challenging for different reasons. First, analysis requires a minimum amount of sediments (e.g. 0.5-1 g), which tends to be samples of 0.5 cm3. In most of the lake records, the sedimentation rate is at mm scale, so annual resolution is not possible. Second, having a continuous record of the Holocene at this resolution is time consuming and expensive.

Nonetheless, the model presented in this study is based on the initial work of Parnell et al. (2015) who built a similar model using pollen. The framework is similar and therefore can be adapted easily to a multivariate biological proxy dataset if a user wishes. For example, new advances in hyperspectral imaging analysis would allow applying this method to a multivariate dataset of biological-related proxies (e.g. pigments, bacterial communities). For this study we choose XRF-CS because it is a state-of-the-art methodology in palaeolimnology and palaeoclimatology, and is relatively fast and cheap, so many other researchers might benefit from this model.

We will add in some more details into the manuscript as we explain the benefits of  $\mu$ XRF-CS and why we chose this data other alternative biological approaches for this study.

Null counts (inferred absences,  $x_i=0$ ) are frequent in most types of proxy datasets, e.g. pollen diagrams. Would this be a problem for your approach if used in other studies, given the clr transformation? Could the approach be amended to account for null counts (even if just by adding a small constant to all values)?

We agree that for many poxy datasets null counts can be a problem. However, as we specify in the manuscript that the data should be clr-transformed prior to using the SCUBIDO model, we would assume that the user's data does not contain many null values as this is a pre-requisite for clr-transformation (Bertrand et al., 2024). However, we thank the reviewer for mentioning this as we will now add some additional information about the pre-requisites of clr-transformation and that this should be addressed before the modelling commences.

Line 161-2, it is unfortunate that age uncertainty is not considered in this modelling approach. This is problematic, even for this varved lake (note that the varves of Diss Mere don't reliably extend to present-day, causing even more chronological uncertainties). Could age uncertainty not be included as a module of the Bayesian analysis? Lots of work has gone into developing Bayesian age-models, so you could build on existing methods (e.g., https://gchron.copernicus.org/articles/4/409/2022/).

We agree with the reviewer that age uncertainty is important to consider. We do envision that future versions of SCUBIDO will incorporate and model age uncertainty. However, for this current version we were unable to find a way to build age modelling and uncertainty quantification at each timestep into the current model without it being very computationally expensive. We really appreciate the reference suggestion, and we will consider using this or other similar models in a future update of SCUBIDO.

Following this comment and a comment from Reviewer 1, we will now add some more information about the chronology into the Supplementary Information and also remind

users throughout that this modelling approach does not include age modelling or account for age uncertainty.

**Lines 163-5, See also Blaauw et al. 2010 who produced random-walk 'fake' proxy datasets, some of them steered by (again fake, random-walk) environmental forcers [https://doi.org/10.1177/095968360935518]**

We appreciate the reviewer's suggestion; it was a very interesting read. However, we are unsure whether the comment refers specifically to the lines stated here (163–165), or to the subsequent lines (166–167) in the previously submitted manuscript, where we describe the random walk component of the model. As this section of the manuscript is focused on outlining the structure of our inference model, we have opted not to include additional references here. If the reviewer feels that the reference to Blaauw et al. (2010) could be more appropriately integrated elsewhere in the manuscript, we would be happy to do so.

**Why did you use a polynomial regression model (equation 4)? Would other models such as a smooth spline have worked?**

In our modelling approach we were interested in finding a balance as we did not want to have a model which was overly complex and computationally demanding, especially since there isn't a hugely complicated relationship between the elements and climate. At the same time, we wanted a model that was simple and fast for users to run, but still capable of capturing the main features of the relationship. Given the relatively weak individual correlations between the elements and climate, we settled on a quadratic relationship as it provided a good compromise between simplicity and performance.

The reviewer is correct that more complicated models could be fitted and a similar example of this modelling approach which includes a more complicated p-spline model was included in Cahill et al., 2023 in *Environmetrics*. We did try a P-spline model as part of our model exploration process, but this tended to overfit the XRF data and drastically reduced the speed of our modelling approach.

**Vague normal/uniform distributions were used as priors. Did you explore how different/stronger priors would affect the results?**

In this modelling approach we have used different types of priors ranging from vague to more informed. The choice behind using a vague prior to, for example, capture the relationship between the elements and climate, was that we did not want to assume that we knew the relationship between the different variables. This was especially important for us as we wanted the model to be easily adaptable to different lakes with significantly different XRF datasets and therefore we did not want to assume the same priors could be applied to all records.

On the other hand, we have also included a more informed prior on the random walk variable for the fossil model as we take this from the modern calibration. We first use a vague prior on the calibration model, but we capitalise on the fact that it is a Bayesian model by borrowing the strength of the modern data where we do know the climate dynamics over time and have applied that into the fossil record. We thank the reviewer

for this comment as it makes us aware that we need to be clearer about the choice of priors in the new version of the manuscript.

Figs. 1/2: I agree with the other reviewer that it is very hard to visually spot much of a correlation between temperature (black dots) and the variables (or between true and reconstructed temperature in Fig. 2 - my eyes would tell me that there's essentially no correlation). This is especially because temperature shows very little variability, ranging only from -2 to 2 degC, with the far majority centered around 0 (so, reconstructed temperature anomalies >1 degC would be based on very few calibration data). But perhaps I am misunderstanding these graphs. Could the individual leave-some-out sets be shown with different colours in Fig. 2?

We agree with all the reviewers that from Figure 1 you cannot see a clear direct relationship between the individual elements and temperature. Please see the response to Reviewer 1 on the reasons why we do not expect to see high correlations.

For Figure 2 after all reviewers identified this, we reassessed the validation code and realised there was a slight bug which was misaligning the reconstructed temperature to the true temperature by one year. This was caused by rounding up to the nearest integer. We have since re-plotted the results and have obtained a much clearer set of validation results.

Can you show the structure of the MCMC run's 'energy'? How many parameters were involved? Line 384, how do you mean 4 chains were used - were the results joined afterward? Was any thinning necessary?

We thank the reviewer for this comment though we are a little unclear as to what is being asked for here. We have used standard MCMC diagnostics to assess convergence of the chains (e.g. the Brooks Gelman Rubin R-hat diagnostic). We also evaluated trace plots of the MCMC chains for many of the parameters to confirm convergence, but for brevity we did not include these in the paper. When we say that four chains are used, we mean that four different sets of starting values were used to run the MCMC in parallel. This is the basis for the R hat test mentioned above. To be clearer for readers we will add these references more times throughout the text.

Figure 4 - the temperature reconstruction of Nautajarvi seems close to what is generally assumed for Holocene temperature time-series, with a HTM followed by a late-Holocene cooling. However this is not visible for Diss Mere. This is reported in the Discussion. A devil's advocate might say that the authors were lucky with Nautajarvi's reconstruction as it follows known Holocene patterns, but weren't as lucky with Diss Mere. Did you try multiple runs with different settings/priors, and were the results robust?

We thank the reviewer for this comment. We did multiple modelling runs, with different starting points and settings, and only observed slight changes, mostly in the variability.

We agree with the reviewer that the reconstruction from Nautajärvi has a HTM whereas Diss Mere is warming through the Holocene with no distinct HTM. However, this is still a big topic in the palaeoclimate field leading to the Holocene temperature conundrum. Discrepancies between proxy-based reconstructions and climate model

simulations suggest that proxy-based reconstructions might be biased toward the summer season as most of these reconstructions are based on biological proxies (Liu et al., 2014). There is also a complex spatial-temporal distribution of the HTM that might explain the regional differences (Cartapanis et al., 2022), e.g. a higher sensitivity to changes in seasonality at high latitudes, different sensitivities in continental vs maritime climate regions. On the other hand, as shown in Figure 5, there are other recent reconstructions that combine proxy data and data assimilation approaches that show no distinct HTM, which agree with the general evolution of temperature recorded in Diss Mere. Nonetheless, a discussion of the HTM is out of the scope of this manuscript, which rather describes a new method to reconstruct temperature.

The manuscript will need a thorough grammar/punctuation check because at times it is difficult to follow. I am suggesting a few specific changes here:

We thank the reviewer for identifying these and adding in the different lines numbers to locate them. We will make all the grammatical errors mentioned.

It is great that the authors have produced an R package for the approach presented here. Unfortunately, I was unable to actually install and run the R package. JAGS has to be installed in order to run. Of the 23 additional packages that had to be installed on my systems (I tried this on both linux and mac), it was 'rjags' that caused issues. This seems to be caused by the version of JAGS (4.3.2) not coming with 'modules-4', or owing to linking problems. Would it be possible to swap to another MCMC sampler? For example, twalk (Christen and Fox 2010 Bayesian Analysis 5: 263-282) is available as an R package. Even though this twalk package uses pure R and will thus be slower than e.g. c++ versions, at least I expect it would be easier for users to install and run everything without too many issues.

We thank the reviewer for bringing this to our attention. We have now highlighted that JAGS is needed to be installed for SCUBIDO to be ran on both its vignette and GitHub pages and we provide a link to download the software.

We agree that perhaps another MCMC sampler would be easier for some users. However, the computational time for this model is very long given the number of data points commonly present in  $\mu$ XRF-CS datasets. Therefore, we have found that using an alternative sampler would not be beneficial. However, we will be monitoring the SCUBIDO package and in the future will be making it available on other platforms, e.g. Python and thus may make it more accessible for some users. We thank the reviewer again for their suggestion but at present we prefer to remain with the JAGS package in R.

---

## Author Comment (AC3)

We would like to thank the anonymous reviewer (Reviewer 3) for their constructive suggestions and comments, which have helped to improve our manuscript entitled "SCUBIDO: a Bayesian modelling approach to reconstruct palaeoclimate from multivariate lake sediment data." Below, the reviewer's comments are shown in red, and our responses in black.

The authors frequently mention quantitative proxy values (say line 78) but are the proxy records they collect not quantitative data, in terms of intensities, as is much of the proxy data collected in terms of tree rings, battles, diatoms, pollen etc. It would be good to clarify this as qualitative proxy records are mentioned throughout but it is not clear to me what this refers to.

We thank the reviewer for this comment, and we agree that this might be confusing. We meant that the climate information derived from proxy data is qualitative but agree with the reviewer that proxy data are quantitative. We will be more specific in the new revised version of the manuscript and will change "qualitative proxy value" (e.g. line 78 in the previous version of the manuscript) for "qualitative climate information derived from proxy values".

(Line 225 and Figure 1). From a modelling perspective there is no clear rationale made for why a quadratic relationship is appropriate, perhaps some form of smoother would work equally well without the tail assumptions of the polynomial model. In Figure 1 visualisations of the fits for each chemical is made however there is no clear relationship and the figures reflect a lot of noise., This may be due to only one climate proxy being presented (temperature?) whereas it is possibly the case that the proxy could depend on several - the authors should comment in this regard.

The reviewer makes some great points here which mirror the responses to the other reviewers. Please see the response to Reviewer 2 about the quadratic relationship. We have tried a P-spline model but found that this significantly underestimated the variability and is very computationally expensive.

We have responded to Reviewer 1 about the lack of a clear relationship between the individual elements and temperature in Figure 1, but as a summary this figure should not view each of the element relationships as independent as it is the joint relationship between all these elements which provides us with the relationship between climate and XRF data used to calibrate.

We completely agree that some of the noise is going to be coming from other meteorological processes and variables. However, this is the reason why we wanted to use a probabilistic approach as there are likely many processes which are involved unrelated to temperature (or another meteorological variable we are reconstructing), and thus we want to account for these within the uncertainties. If we were to have other bits of information that we can include, such as precipitation etc, then it is likely that the uncertainties would be reduced. However, it is challenging to know what those other drivers are, and it is also challenging to get data for this which is of good enough quality to put into the model and have enough computational power to fit within a reasonable timeframe. We thank the reviewer for this for this suggestion though as these are some very interesting points, but at present this is something that we are not able to here. We will add this as a potential avenue for future model updates.

The authors mention that uncertainty quantification is a strong basis of their approach and note (Line 417) "Nevertheless, gaining an 80% coverage percentage is acceptable for this modelling approach ". It appears that the constructed 95% HPD regions only contain 80% of observations. Why is the 80% acceptable, or a stronger argument needs to be made in this regard. Perhaps the reduction in uncertainty coverage is due to the log-ratio transformations of the XRF data and modelling inter-element relationships with a multivariate random effect which is acting as a poor equivalent to accounting for the compositional nature of the data? While the authors claim reasonable performance from an uncertainty quantification perspective, insufficient discussion is made of mean/median (unclear which) predictions in Figure 2. I note that authors later calculate the correlation between Diss Mere and the LMR but provide no similar calculation here? Figure 2 seems to suggest that the R^2 in this case would be very poor, is this why it is not presented?

We thank the reviewer for this comment. We have since re-run the model for Diss Mere and started the calibration period at 1700 CE rather than 1659 CE due to higher uncertainties in the instrumental data and missing years in the XRF data. In addition, after carefully reviewing the validation code we identified a small bug which was rounding up the age variable to the nearest integer. This shifted the predicted temperature by one year and therefore was not directly comparing to the same year in the validation dataset. However, since accounting for this misalignment, we now have a much stronger relationship between true and predicted temperature, and a very good coverage percentage (97.4% of the reconstructed values fit within the 95% confidence intervals). We thank the reviewer for the suggestion about adding in the $r^2$ values, we think this is a great idea and we will add these into the figure. Based on this comment we will also add in a few sentences talking about the median value and how they do not perfectly align and potential reasons as to why.

We have included the new updated true vs reconstructed figures to the response to Reviewer 1.

A weakness (in my opinion) of the results presented in the two case studies later on stems from an insufficient evaluation of predictions of the mean temperature for the calibration dataset at Diss Mere - predictive performance in terms of the uncertainty may be 80% coverage, but it is not clear that the center of the prediction intervals are accurate - this perhaps explains the commentary on the performance of Diss Mere later on. Was an analysis of the calibration approach carried out at the Finnish site? If so, does it explain why the predictive performance is potentially better there than Diss Mere? Alternatively, do the weaknesses in predictive performance in Figure 2 also manifest at the Finnish site? Similarly annual mean temperature is used - is precipitation potentially useful to incorporate here or is predictive performance poor when it is incorporated as observed at Diss Mere? Additional evaluations in this regard could be included in the Supplementary materials.

We thank the reviewer for this, and we hope that the response to your previous comment has addressed the concerns about the predictive performance. We think it is a very good idea to add in the validation information for Nautajärvi, and we will include this as part of the supplementary information as a response to this comment. Please see Figure A1-4 in the response to Reviewer 1 for the Nautajärvi figure.

In terms of model assessment, the sensitivity of results to specified priors is not provided - the priors as presented are very vague but could some information be incorporated to make these more informative? Since the XRF is rescaled using a centred log ratio - is it plausible that intercept values of +-200 are possible, which is what is suggested by the vague prior. Similar arguments apply to the priors for the other components - are some of the values suggested by the prior impossible?

We are grateful to the reviewer for pointing this out. However, we think there might be a misunderstanding of our probability distributions. We are using this standard nomenclature of a normal distribution being represented by its mean and variance. Thus, when we write N (0, 100) we are referring to a normal distribution with mean zero and standard deviation 10 (variance 100). Thus, the intercept has an a priori 95% range of -20 to 20 which seems plausible for CLR transformed data. This is still a vague prior, but we would argue that it is weakly informative for the scales of the data that we are modelling.

Please refer to the response to Reviewer 2 about or choice behind the different types of priors used within this model.

The manuscript also requires substantial editing as there are a number of grammatical errors, typos and excessive use of language which makes the manuscript difficult to follow at times. For eg, SCUBIDO is spelt incorrectly twice and some of the text used either side of equations causes confusion. I have noted several of these below.

We thank the reviewer for identifying these and we will change all of these in the new version of the manuscript.

Line 309 - "found  a qualitative link" - what is a qualitative link?

We suggested that there was a qualitative link as there was a good visual relationship between Holocene temperature evolution and the Ca record. We will explain this in more detail.

Line 334 - "and thus the model did not find a good enough relationship. Annual mean temperature on the other hand worked well, which support the temperature signal recorded in the qualitative XRF-CS data during the Holocene " - what is meant by a "good enough relationship? Why was the temperature signal and XRF-CS relationship deemed good enough?

We thank the reviewer for identifying that more detail is needed here. We will explain in more detail why precipitation was not used as the reconstruction was flat, there was no predictive power between the elements and precipitation and the validation showed no relationship between true and reconstructed precipitation.

Line 337: "Another point to highlight at this stage is that we run the Bayesian model using a   multivariate dataset made of the elements measured by the XRF scanner, which differentiates SCUBIDO from other recent reconstructions based on varved sediments " - How does it differ?

We were referring to other approaches using only single elements to infer climate, or a pair of elements in the form of a ratio. However, we will now remove this sentence following the suggestion from Reviewer 1 as we mention this later in the manuscript.

Line 339: "We therefore rely on the Hadley Central England Temperature (HadCET, Met Office) data" - is this proximate to the site?" As such, does it capture temperature change at the site reasonably?

Unfortunately, we could not find another station which is closer to the site and is also long enough to calibrate the data as the top of the XRF record is at 1932. We will add some additional information in the text about this in response to this comment.

Line 350: "$XRFm$ was resampled to annual means " - How was it sampled or adapted? Was the XRF data not at annual level in any case?

We used linear interpolation to downscale our resolution form many data points per year to one. We will clarify this in the text based on this comment and the comment from Reviewer 1.